# Antihydrogen accumulation for fundamental symmetry tests

M. Ahmadi[1], B.X.R. Alves[2], C.J. Baker[3], W. Bertsche[4,5], E. Butler [6], A. Capra[7], C. Carruth[8], C.L. Cesar[9], M. Charlton[3], S. Cohen[10], R. Collister[7], S. Eriksson[3], A. Evans[11], N. Evetts[12], J. Fajans [8], T. Friesen[2], M.C. Fujiwara[7], D.R. Gill[7], A. Gutierrez[13], J.S. Hangst[2], W.N. Hardy[12], M.E. Hayden[14], C.A. Isaac[3], A. Ishida[15], M.A. Johnson[4,5], S.A. Jones[3], S. Jonsell [16], L. Kurchaninov[7], N. Madsen [3], M. Mathers[17], D. Maxwell[3], J.T.K. McKenna[7], S. Menary[17], J.M. Michan[7,18], T. Momose[12], J.J. Munich[14], P. Nolan[1], K. Olchanski[7], A. Olin [7,19], P. Pusa[1], C.Ø. Rasmussen[2], F. Robicheaux [20], R.L. Sacramento[9], M. Sameed[3], E. Sarid[21], D.M. Silveira[9], S. Stracka[22], G. Stutter [2], C. So[11], T.D. Tharp[23], J.E. Thompson[17], R.I. Thompson[11], D.P. van der Werf[3,24] & J.S. Wurtele[8]

Antihydrogen, a positron bound to an antiproton, is the simplest anti-atom. Its structure and properties are expected to mirror those of the hydrogen atom. Prospects for precision comparisons of the two, as tests of fundamental symmetries, are driving a vibrant programme of research. In this regard, a limiting factor in most experiments is the availability of large numbers of cold ground state antihydrogen atoms. Here, we describe how an improved synthesis process results in a maximum rate of $10.5 \pm 0.6$ atoms trapped and detected per cycle, corresponding to more than an order of magnitude improvement over previous work. Additionally, we demonstrate how detailed control of electron, positron and antiproton plasmas enables repeated formation and trapping of antihydrogen atoms, with the simultaneous retention of atoms produced in previous cycles. We report a record of 54 detected annihilation events from a single release of the trapped anti-atoms accumulated from five consecutive cycles.

[1] Department of Physics, University of Liverpool, Liverpool L69 7ZE, UK. [2] Department of Physics and Astronomy, Aarhus University, DK-8000 Aarhus C, Denmark. [3] Department of Physics, College of Science, Swansea University, Swansea SA2 8PP, UK. [4] School of Physics and Astronomy, University of Manchester, Manchester M12 9PL, UK. [5] Cockcroft Institute, Sci-Tech Daresbury, Warrington WA4 4AD, UK. [6] Physics Department, CERN, CH-1211 Geneve 23, Switzerland. [7] TRIUMF, 4004 Wesbrook Mall, Vancouver, BC, Canada V6T 2A3. [8] Department of Physics, University of California at Berkeley, Berkeley, CA 94720-7300, USA. [9] Instituto de Fisica, Universidade Federal do Rio de Janeiro, Rio de Janeiro 21941-972, Brazil. [10] Department of Physics, Ben-Gurion University of the Negev, Beer-Sheva 84105, Israel. [11] Department of Physics and Astronomy, University of Calgary, Calgary, AB, Canada T2N 1N4. [12] Department of Physics and Astronomy, University of British Columbia, Vancouver, BC, Canada V6T 1Z1. [13] Department of Medical Physics and Biomedical Engineering, University College London, London WC1E 6BT, UK. [14] Department of Physics, Simon Fraser University, Burnaby, BC, Canada V5A 1S6. [15] Department of Physics, The University of Tokyo, 7-3-1 Hongo, Tokyo 113-0033, Japan. [16] Department of Physics, Stockholm University, SE-10691 Stockholm, Sweden. [17] Department of Physics and Astronomy, York University, Toronto, ON, Canada M3J 1P3. [18] École Polytechnique Fédérale de Lausanne (EPFL), Swiss Plasma Center (SPC), CH-1015 Lausanne, Switzerland. [19] Department of Physics and Astronomy, University of Victoria, Victoria, BC, Canada V8P 5C2. [20] Department of Physics and Astronomy, Purdue University, West Lafayette, IN 47907, USA. [21] Soreq NRC, Yavne 81800, Israel. [22] Universita di Pisa and Sezione INFN di Pisa, Largo Pontecorvo 3, 56127 Pisa, Italy. [23] Physics Department, Marquette University, P.O. Box 1881, Milwaukee, WI 53201-1881, USA. [24] IRFU, CEA/Saclay, F-91191 Gif-sur-Yvette, France. Correspondence and requests for materials should be addressed to T.F. (email: Tim.Friesen@cern.ch) or to N.M. (email: Niels.Madsen@cern.ch)

Enormous progress in antihydrogen ($\overline{\mathrm{H}}$) synthesis and trapping has been made in recent years[1–3] and transitions between internal states have been induced and observed[4, 5]. While the results of measurements conducted to date are consistent with the charge-parity-time invariance theorem, the field is still very much in its infancy. Significant work will be needed to achieve the levels of measurement precision obtained in the study of matter atoms. Ultimately, precision studies of antihydrogen properties complement, and are complemented by, experiments that probe the foundations of the standard model including studies of antiprotons[6], antiprotonic Helium[7], muonic atoms[8] and positronium, the electron-positron-bound state[9].

Antihydrogen is synthesised from antiprotons ($\overline{\mathrm{p}}$) and positrons ($e^+$); the most widely employed method starts with cold plasmas of both species, which are brought together in Penning–Malmberg traps, where axial magnetic fields provide radial confinement and electric fields provide axial confinement (see ref. [10] for a review). Antihydrogen atoms in states where the magnetic moment is anti-aligned with the magnetic field are then confined in a magnetic minimum trap, provided their kinetic energy is low enough[1–3]. This is typically <0.5 K in temperature units for Tesla-scale trapping fields (multiplication by the Boltzmann constant $k_{\mathrm{B}}$ to obtain energy units is implicit). Trapped antihydrogen is detected by ramping down the currents in the magnetic trap over 1.5 s and detecting the annihilation of the antiproton when the released atoms hit the wall of the trap. We employ a three-layer silicon vertex detector[11] to image the annihilation vertex position of each detected atom. Event topology is used to distinguish antiproton annihilations from cosmic rays.

Here we report a breakthrough in the efficiency of antihydrogen trapping and a method for accumulating or stacking anti-atoms trapped during consecutive production cycles. These advances are realised through the development of a number of techniques that yield both more and colder antihydrogen. Improved methods to produce cold antihydrogen are critically important to most experimental initiatives in the field, and hence the results presented here are of broad relevance; see refs. [3, 10, 12] for examples.

## Results

**Antiproton and positron preparation.** The ALPHA apparatus comprises three systems that allow antiproton capture, positron accumulation and antihydrogen synthesis. The central apparatus in which antihydrogen is formed is called ALPHA-2. It has been designed to allow the overlap of laser light and microwaves with trapped antihydrogen; a schematic view of the device is shown in Fig. 1. Antiprotons from the CERN antiproton decelerator[13] (AD) are captured[14] in a high voltage (4 kV) Penning–Malmberg trap that we refer to as the catching trap (CT, not shown). Once captured they are sympathetically cooled by a batch of pre-loaded electrons which self-cool by the emission of cyclotron radiation in the cryogenic (~6 K) environment of the trap. For this purpose we use a large (radius ~9 mm) plasma comprising ~$8.5 \times 10^7$ electrons, which reduces the kinetic energy of about 73% of the antiprotons to below 100 eV in 20 s (this efficiency is not a fundamental limitation[15]; it is merely a practical compromise adopted to realise the demonstrations reported here, without excessive fine-tuning of electron plasma and sequence timing parameters). Any uncooled antiprotons are subsequently ejected by reducing the depth of the potential well. To secure an efficient transfer to the ALPHA-2 apparatus, the combined electron–antiproton plasma is then radially compressed using the rotating wall technique[15] in the strong drive regime[16]. In this regime, the rotating wall achieves a plasma density proportional to the applied frequency up to a maximum. Since the radial extent of the plasma is limited by this maximum, fewer electrons allow for a smaller final size. Prior to applying the rotating wall most of the electrons used for cooling are therefore ejected; ~$10^6$ are retained. After compression, the antiprotons are allowed to re-cool for ~10 s before the remaining electrons are ejected. This protocol, which takes 100 s to complete, typically yields a plasma of $1.1 \times 10^5$ antiprotons with a radius of 0.2 mm at ~400 K. These antiprotons are then ejected from the CT with 25 eV of kinetic

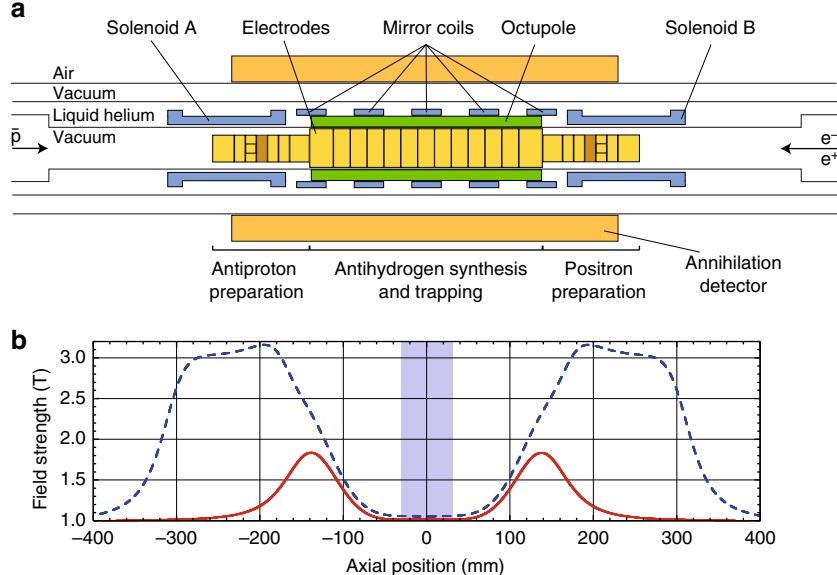

**Fig. 1** The ALPHA-2 central apparatus. **a** ALPHA-2 geometry, drawn to scale except for the radial extent of the annihilation detector. The inner diameter of the Penning–Malmberg electrodes is 44.35 mm in the central region of the atom trap and 29.6 mm at either end. Antiprotons enter from the *left* in this view, while positrons and electrons are loaded from the *right*. **b** Magnetic field strength on axis with the atom trap energised (the external solenoid responsible for producing a uniform 1 T field is not shown). The *solid curve* (*red*) shows the flattened atom trap field used in ref. [5]. The *dashed curve* (*blue*), shows the on-axis field during stacking; the *left* and *right* solenoids **a**, **b** increase the field from 1 to 3 T for enhanced capture, cyclotron cooling and rotating wall efficiency of, as appropriate, positrons, electrons and antiprotons

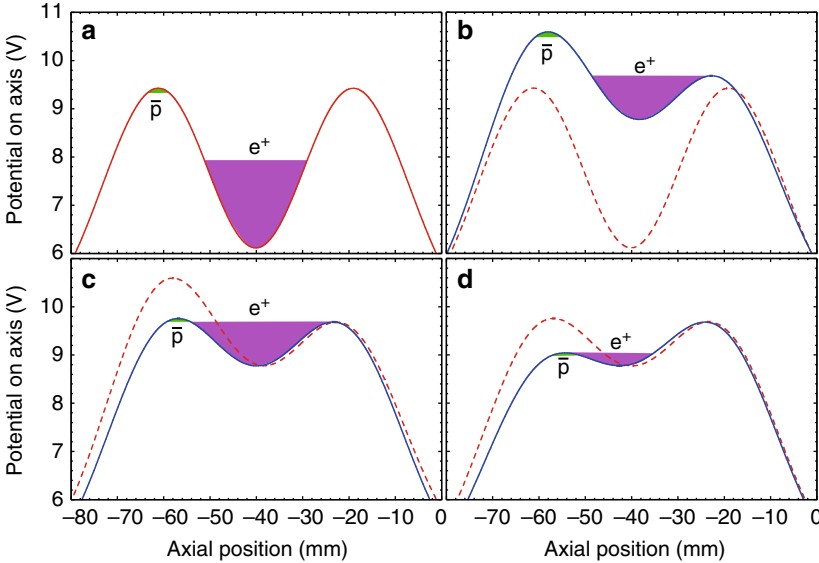

**Fig. 2** Antihydrogen synthesis sequence. *Dashed* and *solid curves* represent electrostatic potentials before and after each step in the process. *Filled regions* indicate self-potentials and physical extents of antiproton and positron plasmas. **a** Potential before evaporative cooling. Positron well depth 3.31 V. **b** Evaporative cooling, during which energetic positrons escape to the right (duration 600 ms). Final positron well depth 0.91 V. **c** Potential realignment in preparation for mixing (duration 10 ms). Final positron well depth 0.91 V. **d** Potential merge mixing (duration 1 s). Positrons escape to the left during mixing, resulting in further evaporative cooling. Final positron well depth 0.27 V. Remaining positrons are ejected to the right for a temperature measurement; remaining antiprotons are ejected to the left

energy by momentarily (~2 µs) dropping the confining potential. Antiproton (and positron) temperatures are obtained from destructive measurements of the axial energy distribution[17–19]. Positron temperatures are measured after each mixing cycle, as part of the process in which charged particles are removed from the trap. Radial distributions are determined by ejecting particles to an externally located multichannel plate/phosphor detector[18].

Antiprotons ejected from the CT move ballistically to the ALPHA-2 apparatus (Fig. 1), guided only by static axial magnetic fields. Upon arrival, they are manipulated in a manner analogous to the treatment they received in the CT described above. The recapture process is accomplished using an electrostatic potential well pre-loaded with electrons and the compression, cooling, electron ejection and re-cooling sequences employed in the CT are repeated without antiproton loss, resulting in a plasma with a radial extent of 0.4 mm and a temperature of ~100 K. This preparation takes about 100 s. These $\bar{p}$ are then evaporatively cooled[19] to 40 K over 10 s to leave around $9 \times 10^4$ of them in a cloud with a radial extent of 1 mm.

In parallel with the antiproton manipulations described above, positrons are prepared in the opposite end of ALPHA-2. The positrons originate from a radioactive $^{22}$Na source that feeds a Surko-type buffer-gas accumulator[20]. Plasmas comprising between $10^6$ and $10^8$ positrons are generated, depending on the details of the experimental sequence, and transferred to ALPHA-2[21]. In order to control the positron number and density, which would otherwise drift (driven by conditions in the accumulator that evolve on a timescale of days), a combination of evaporative cooling and strong drive rotating wall compression is now used[22]; this protocol reduces variations to <1 %. Subsequently the positrons are left to thermalise with their surroundings in a deep (117 V) well before eventually being transported to a shallow (3.31 V; Fig. 2a) well in the mixing region (see below). The increase in plasma length associated with the transfer to a shallow well causes the positron plasma to cool adiabatically to about 30 K (cf. ~50 K if this step is omitted)[23].

The key drivers underlying the positron and antiproton preparation protocols outlined above can be elucidated as follows; first, overlap during mixing is optimised by ensuring that the $\bar{p}$ and $e^+$ plasma radii are similar. Second, plasma temperatures as low as achievable are targeted, since this results in improvements in both synthesis and trapping. Finally, small plasmas are prepared to avoid deleterious influence of the octupole field of the magnetic minimum trap. Further discussion of this last point will follow when we describe the stacking procedure.

**Antihydrogen synthesis.** Before the final preparation and mixing steps, the magnetic minimum trap is energised. The sequence of electrostatic potentials used in these final stages is shown in Fig. 2. Following the adiabatic cooling step (potential not shown), the positrons are evaporatively cooled (Fig. 2a, b) by lowering the potential barrier on the right from 3.31 to 0.91 V over 600 ms (the electrostatic self-potential of the positrons is initially about 1.8 V). At this point, the positrons start to re-thermalise with their surroundings, and so the potentials are quickly (in 10 ms; Fig. 2c) modified to the point where the antiprotons are on the verge of entering the positron plasma. Finally, the antiprotons and positrons are merged by lowering the potential barrier between them over about 1 s; during this process antiprotons are free to enter the positron plasma and positrons are free to drift to the left (Fig. 2d). While it may seem counter-intuitive to intentionally release positrons from the trap when synthesising antihydrogen, this technique offers two advantages. The potential difference between the antiprotons and the positrons is minimised without accelerating the former, while the latter are continuously cooled via evaporation during the merging process. This effect helps check the heating we observe when the positrons are held in a static well after evaporative cooling. A significant reduction in the post-mixing positron temperature is observed relative to the autoresonant (AR) antiproton injection technique employed previously, in agreement with simulations[24]. As shown in Table 1, typical temperatures after AR injection mixing were 50–70 K, whereas temperatures after the potential merge mixing process are in the range 15–20 K. The AR process resulted in heating of the positron plasma that could only be reduced by decreasing the

**Table 1 Figures of merit characterising antihydrogen formation and trapping efficiencies**

| p̄ injected (×10⁴) | H̄ formed (×10⁴) | $T_{e+}$ before (K) | $T_{e+}$ after (K) | Number of trials | H̄ trapped & detected | H̄ trapping efficiency (× 10⁻⁴) |
|---|---|---|---|---|---|---|
| *AR injection mixing*[a] | | | | | | |
| 1.31 ± 0.01 | 0.53 ± 0.01 | 27 ± 3 | 51 ± 1 | 54 | 0.62 ± 0.11 | 1.6 ± 0.3 |
| 3.1 ± 0.1 | 0.87 ± 0.01 | 27 ± 3 | 60 ± 1 | 27 | 0.59 ± 0.15 | 0.9 ± 0.2 |
| *Potential merge mixing*[b] | | | | | | |
| 5.5 ± 0.1 | 2.4 ± 0.1 | 18 ± 2 | 16 ± 1 | 16 | 8.7 ± 0.7 | 4.7 ± 0.4 |
| 9.0 ± 0.3 | 3.1 ± 0.1 | 18 ± 2 | 17 ± 1 | 26 | 10.5 ± 0.6 | 4.7 ± 0.3 |

Antiprotons were evaporatively cooled to ~40 K in all cases. [a]AR injection mixing. The positron plasma comprised 2.3 × 10⁶ positrons at a density of 1.3 × 10⁸ cm⁻³ and a radius of 0.55 mm. [b]1 s potential merge mixing. The positron plasma comprised 1.6 × 10⁶ positrons at a density of 6.5 × 10⁷ cm⁻³ and a radius of 0.66 mm. The trapping efficiency is the number of trapped antihydrogen divided by the number formed (annihilation detector efficiencies included). The positron densities were set by tuning the evaporative cooling process to achieve maximum trapping efficiency. Uncertainties are statistical and assume the parent distributions are Poissonian

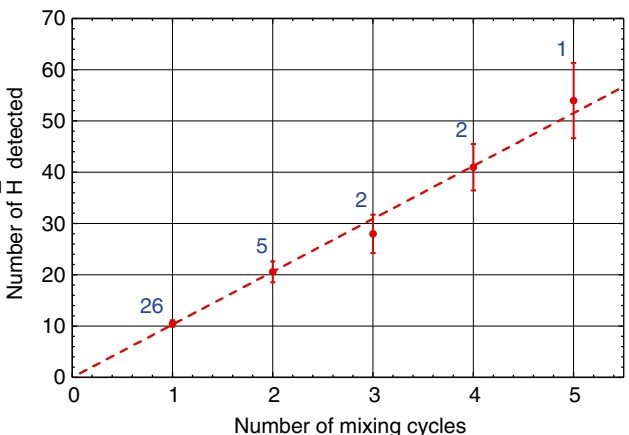

**Fig. 3** Antihydrogen stacking. The number of antihydrogen atoms detected when the magnetic minimum trap is ramped down after one or more consecutive mixing cycles. Each mixing cycle in a sequence is separated by ~4 min. The *error bars* are statistical and the number of replicates is indicated above each data point. The *dashed line* is a linear fit to the data giving an average trapping rate of 10.5 ± 0.6 detected antihydrogen atoms per mixing cycle

number of injected antiprotons, which unfortunately lowered the antihydrogen yield. By contrast, the potential merge process exhibits an antihydrogen formation rate that increases with the number of antiprotons injected, with no adverse temperature effects.

Table 1 compares variations in several key attributes of the two mixing procedures as the number of injected antiprotons is increased. Lower positron temperatures are evident when the potential merge mixing protocol is used. Doubling the number of antiprotons used in AR injection mixing increases the number of antihydrogen formed (Table 1), but simultaneously heats the positrons and ultimately reduces the trapping efficiency. Conversely, starting from a significantly larger initial value, and then doubling the number of antiprotons employed in the potential merge process increases antihydrogen trapping and reveals no measurable change in the positron temperature (Table 1). In fact, the positron temperature remains unchanged during potential merge mixing due to evaporative cooling.

The trapping fraction, while much improved for the potential merge process (likely because of the lower positron temperatures) is a factor of ~3 below what one might expect solely on the basis of the fraction of atoms in a Maxwell–Boltzmann distribution that have energies less than the trap depth (0.54 K in temperature units), which for 20 K H̄ is 1.6 × 10⁻³. This assumes atoms form in trappable and un-trappable states with equal probability, when

in fact theoretical work[25–27] suggests they are more likely to form in un-trappable states. It also assumes that the antiprotons equilibrate with the positrons before antihydrogen is formed[10, 28]. If instead the initial antiproton temperature (~40 K) carries over to the H̄, the discrepancy is almost eliminated. We note that the improvement in trapping efficiency is roughly in accord with expectations for lower positron temperatures[10, 28]. It is likely the positron density also plays a role in determining the trapping rate. However, because of the inter-dependence between temperature and density imposed by the application of evaporative cooling[19], detailed studies of trapping rate as a function of either parameter are still pending. For now, we note that the record trapping rates we have achieved using potential merge mixing yielded about an order of magnitude more trapped atoms in a single 6-month experimental run than were accumulated over several years using AR injection mixing.

**Antihydrogen accumulation.** A batch of antiprotons is delivered by the antiproton decelerator roughly every 2 min. The typical preparation time for antihydrogen synthesis as described above is 3–4 min, and experiments with trapped antihydrogen can take up to 20 min. This scenario creates an important resource efficiency and productivity challenge, whose goal is to optimise the number of antihydrogen atoms that are employed in scientific experiments. Additionally, having more trapped atoms present during long exposures to laser or microwave radiation improves the signal-to-background ratio for observation of H̄ annihilations. Building on the advances in plasma control and trapping efficiency described above, we have, therefore, implemented a new experimental protocol that increases the availability of trapped antihydrogen for experiment by making use of a larger fraction of the antiproton pulses that are delivered and the number of antihydrogen atoms available in each trial. We do this by stacking anti-atoms produced in consecutive mixing cycles.

In the absence of the transverse trapping fields, charged particles are constrained by the strong, uniform magnetic field of the Penning–Malmberg trap to move on the magnetic field lines near the axis of our apparatus. When the transverse octupole trapping field is superimposed on the uniform field, the field lines are distorted and some will ultimately guide particles to the wall as they drift in the axial direction. This imposes a dynamic aperture, parameterised by a critical radius, which is the radial limit beyond which loss is incurred. The critical radius for particles passing through our ~30 cm long fully energised neutral trap is about 4.5 mm[29, 30]. Motivated by this, the electron, positron and antiproton plasmas we prepare have radii of <1 mm, well below this limit.

We have implemented a protocol in which we maintain the magnetic minimum trap fields energised during preparation of the charged particle plasmas. This allows us to produce more

antihydrogen while the atoms from a previous synthesis cycle remain trapped. No detrimental effects on plasma preparation were observed, consistent with the dynamic aperture considerations above. We were able to repeat the antihydrogen synthesis process about every 4 min with the magnetic minimum trap on. This antihydrogen stacking procedure was repeated for a maximum of 54 min cycles, limited by thermal considerations in the octupole current supply circuit; hence the first batch of antihydrogen was confined in the apparatus for 16 min. Figure 3 shows the resulting average number of trapped and detected anti-atoms as a function of the number of cycles. The five-cycle experiment yielded 54 detected antihydrogen atoms, with the detection efficiency of $73.0 \pm 0.4\%$ implying that about 74 atoms were actually trapped. Here we use the analysis procedure described in ref. [5] where an improved signal-to-background ratio for observed antihydrogen was achieved by re-tuning hit thresholds and track fitting in vertex reconstruction. The background is negligible at about $0.063 \pm 0.003$ false positives per trial. We observe a linear increase in the number of trapped atoms with the number of stacks, consistent with no loss of antihydrogen and an average accumulation rate of $10.5 \pm 0.6$ detected $\overline{H}$ per mixing cycle, or a maximum absolute effective accumulation rate of $2.6 \pm 0.2$ detected $\overline{H}$ per minute.

## Discussion

We have demonstrated how detailed plasma control, lower positron temperatures and potential merge mixing of cold $e^+$ and $\overline{p}$ plasmas has enabled a significant increase in the rate and efficiency of antihydrogen trapping. A key feature of the potential merge mixing protocol is that it allows more antiprotons to be used than previously possible. Furthermore, these developments facilitated the implementation of repeated synthesis cycles contributing to the same batch of trapped antihydrogen, to allow several 10s of anti-atoms to be held simultaneously in the trap. These techniques were key for the recent first observation of the 1S-2S transition in antihydrogen[5] and they will almost certainly enhance the precision of any experiment limited by the availability of cold trapped antihydrogen. Importantly, nearly all of these methodologies are equally applicable to experiments that require antihydrogen, but not trapping. The observed strong influence of positron temperature on both antihydrogen formation rate and trapping efficiency indicates that further effort to lower positron temperatures will be well rewarded (e.g., ref. [28]). As a final note, it is encouraging that the potential merge mixing technique appears, thus far, to have no upper limit as to the number of antiprotons that may be exploited, thus paving the way for efficient use of the new ELENA facility that is being constructed at CERN[31]. ELENA promises up to a factor 100 increase in the number of trapped antiprotons available to experiments. All of these developments are encouraging for future fundamental tests with antihydrogen.

**Data availability**. The data sets generated during and/or analysed during the current study are available from N.M. and J.S.H. (niels.madsen@cern.ch, jeffrey.hangst@cern.ch) on reasonable request.

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

## Acknowledgements

This work was supported by the European Research Council through its Advanced Grant programme (J.S.H.); CNPq, FAPERJ, RENAFAE (Brazil); NSERC, NRC/TRIUMF, EHPDS/EHDRS, FQRNT (Canada); FNU (NICE Centre), Carlsberg Foundation (Denmark); JSPS Postdoctoral Fellowships for Research Abroad (Japan); ISF (Israel); STFC, EPSRC, the Royal Society and the Leverhulme Trust (UK); DOE, NSF (USA); and VR (Sweden). We are grateful for the efforts of the CERN AD team, without which these experiments could not have taken place. We thank Jacky Tonoli (CERN) and his staff for extensive, time-critical help with machining work. We thank the staff of the Superconducting Magnet Division at Brookhaven National Laboratory for collaboration and fabrication of the trapping magnets. We thank C. Marshall (TRIUMF) for his work on the ALPHA-2 cryostat. We thank Professor F. Besenbacher (Aarhus) for timely support in procuring the ALPHA-2 external solenoid. This experiment was based on data collected using the ALPHA-2 antihydrogen trapping apparatus. The ALPHA-2 apparatus was designed and constructed by the ALPHA Collaboration using methods developed by

the entire collaboration. All authors contributed to this work as members of the ALPHA antihydrogen collaboration.

## Author contributions

All authors participated in the operation of the experiment and the data taking activities. The experimental protocols were conceived by B.X.R.A., E.B., W.B., C.C., A.E., J.F., A.G., A.I., S.A.J., T.F., N.M., D.M., M.S., D.M.S. and C.Ø.R. Detailed analysis of the data was done by J.T.K.M. and N.M. The manuscript was written by N.M. with assistance from M.E.H. and M.C. The manuscript was then edited and improved by the entire collaboration.

## Additional information

**Competing interests:** The authors declare no competing financial interests.

