## [Peer review file · Nature Communications]

Reviewers' comments:

Reviewer #1 (Remarks to the Author):

This is an excellent manuscript in all regards. I recommend publication in its present form, save for two minor suggestions described below.

What are the major claims of the paper?

The authors have discovered much improved methods to merge positron and antiproton plasmas to form trappable antihydrogen. Not only is the trapping efficiency improved, but trapped antihydrogen pulses can be "stacked," Thus, subsequent measurements on them can benefit from larger numbers of detected antiprotons and hence increased signal to noise relative to using the individual pulses serially.

Are they novel and will they be of interest to others in the community and the wider field?

These new techniques include several substantive new methods and protocols. The increased trappable antiatom production rate is increased about an order of magnitude. This is a huge improvement over previous work. Many groups are either studying now, or plan to study, antihydrogen. The present work should be of considerable interest to those focusing on antihydrogen, and more generally, to those interested in improved techniques to create tailored collections of antiparticles.

Is the work convincing, and if not, what further evidence would be required to strengthen the conclusions?

The work appears to be carefully done and the manuscript is well written.

On a more subjective note, do you feel that the paper will influence thinking in the field?

The answer is definitely "yes."

Please feel free to raise any further questions and concerns about the paper.

Two minor comments. In Line 207, I have trouble determining what "the latter" refers to. It might be good to rephrase this passage. The sentence on lines 213 - 218 would benefit from rewriting also.

We would also be grateful if you could comment on the appropriateness and validity of any statistical analysis, as well the ability of a researcher to reproduce the work, given the level of detail provided.

1. The statistical analysis is good.
2. The paper makes clear what the new procedures are, and so it is certainly adequate in that regard. As mentioned above, the techniques described will, for example, be of rather direct use to the other antihydrogen groups at CERN.

Reviewer #2 (Remarks to the Author):

Dear Editor,

the manuscript

Antihydrogen accumulation for fundamental symmetry tests
by the ALPHA collaboration of CERN reports on two new achievements in antihydrogen physics,

- 1.) a tenfold improved antihydrogen trapping rate, compared to earlier experiments, and
- 2.) the first demonstration of stacking of antihydrogen atoms and the report of the detection of 54 antihydrogen annihilations from a single trap release.

Both achievements together constitute a significant and very important improvement in antihydrogen research and are crucial ingredients to reach the final goals of this branch of fundamental experimental physics – the high-precision comparisons of the properties of hydrogen and antihydrogen to test the fundamental CPT invariance of the Standard Model of Particle Physics.

My overall impression on this manuscript is very positive and given the described results, which considerably contributed to the recent great success of the ALPHA collaboration, publication in Nature Communications can be considered after revision.

The manuscript details procedures which were applied to increase the antihydrogen trapping rate in ALPHA. An important aspect which is missing, is the discussion of the statistical stability of the results. It is obvious, e.g. from the linear scaling of the stacking experiments, that the stability of the experimental sequences meets the requirements which allow the conclusions made in the manuscript, but the discussion of the run-stability should be much more explicit. E.g. just by giving the number of runs which contribute to the results shown in Tab. 1, together with the errors and the explicit statement that the given error is based on the assumption that the results come from a specified parent distribution would already be convincing.

A second point which I would like to address is that the details of the determination of many parameters which enter this manuscript are not given. One important example is the positron temperature, a parameter which is used in the qualitative interpretation of the data. I'm aware that plasma temperature measurements are described elsewhere, but given the importance of positron temperature for this manuscript, it would be adequate to briefly comment for example: "Positron temperature measurements are described in [refs]. For the temperature values used in this manuscript we apply the following method...[brief but explicit description of the experiment sequence]... "

One of the major data sets which is presented in this paper is summarized in Tab 1. As argued above, a column should be added which displays the number of runs taken for each parameter-set. All the parameters contain errors except the amount of injected pbars, the authors may want to comment on that or may want to add an error to the pbar number. The table caption summarizes the positron densities, obviously different densities were chosen for the AR experiments and the potential merge experiments. Naively I would have expected that higher positron densities would lead to even higher hbar production rates. Can the authors please comment why they have chosen lower positron densities in the potential merge schemes? How does this parameter change/affect the direct comparison of the two schemes? Is there any theoretical model which would consistently describe the AR and the PM dataset just by changing positron density and positron temperature? No problem if not, but then the authors should add a short comment. It is not necessarily required, but if the results

could be presented in a more theoretical rather than a purely qualitative framework, it would add additional quality to the manuscript.

Throughout the description of the experiment sequences detailed time information is missing. Only P6/L226 summarizes that the above described preparation sequence takes about 4 minutes. The authors should provide more detailed timing information along the entire description of the experiment sequences and I even propose to add a figure which illustrates the timing / parallelization sequence. This will support the readability of the manuscript. How long are e.g. compression times, ramp times for evaporative and adiabatic cooling etc..? I'm aware that these results were partly published elsewhere, but to make this manuscript more self-consistent, it would be important to describe the exact sequence which was applied to produce these results in a more detailed way.

Once these comments will be addressed in a revised version of the manuscript, I will recommend publication in NC.

Detailed Comments:

Page 2, Line 63 ff:

For a journal like NC, I propose that the authors provide a broader introduction. Rather than just describing CPT tests with antihydrogen also other CPT-testing efforts (antiprotons / antiprotonic Helium / muons / electron-positron) should be summarized/cited in one or two sentences.

Page 3, Line 112 ff:

Description of pbar cooling by electrons: "Most" of the electrons are ejected before RW cooling, and afterwards the remaining fraction of electrons is removed. I'd like to ask the authors to detail: Which fraction of electrons is initially removed, and how long is the cooling time after RW compression before the remaining electrons are ejected? The quantification of the cycle time of such a preparation cycle would be interesting.

Page 3, Line 126 ff:

Description of re-preparation of pbars in ALPHA2: Please add also here time information. How long does the re-preparation take, and how long are your evaporative cooling ramps?

Page 4, line 140 ff:

"which can vary from the accumulator"

About which variations are we talking here? 1%, 10% or even 50% to 90%? Please specify. If possible, please indicate how this variation affects the production/trapping rate of hbars, and how the fluctuations influence the scatter given in Tab 1.

Page 4 / Page 5, Lines 173 to 187:

Very interesting and nice description. Although already published elsewhere, it would improve the self-consistency and readability of this article and add one or two sentences which describe how positron temperature is measured. Please comment on the significance of the temperature measurements. E.g. if the amount of positrons during the potential merge scheme is reduced, how is their temperature determined after a mixing process? Or is this temperature information based on offline experiments without antiprotons, or is a "leak rate" measured online while the potential merge scheme is performed?

Citations:

Page 2, Line 91: "All experimental initiatives in this field..." -> other antihydrogen collaborations could be cited.

Reviewer #3 (Remarks to the Author):

The article "Antihydrogen accumulation for fundamental symmetry tests" by Ahmadi et al (ALPHA Collaboration) reports on several advances in techniques useful for antihydrogen production and trapping, which in summary lead to a tenfold increase of the antihydrogen trapping efficiency. This work is very useful in the context of upcoming precision spectroscopy of antihydrogen, as a test of the fundamental principles of Nature.

Among the reported improvements, two are particularly noteworthy. The "potential merge mixing" of the positron and antiproton plasmas yields, somewhat counterintuitively, significantly more, and also colder antihydrogen. Its yield is also scalable by increasing the number of trapped antiprotons. Thus this new technique will be of prime importance to exploit the 100x larger antiproton plasmas expected at the new ELENA facility.

The second new technique is the demonstration of "stacking" of antihydrogen atoms from several successive mixing cycles. A linear scaling in the number of detected antiprotons is observed, paving the way to high precision high signal-to-background spectroscopy.

The manuscript is very well written and the few comments I have are really miniscule:

line 77: "low enough" could be quantified, (0.5 K for the 1 Tesla field employed). This number is given only on page 5 (line 205)

line 105: A word on why "only" 73% of the antiprotons are cooled may be in order. What prevents the remaining ones from being cooled?

line 106,118: The authors switch between eV and K. The broader readership would probably benefit from mentioning a conversion factor or giving once both the energy and the temperature.

Reviewer #1 (Remarks to the Author):

This is an excellent manuscript in all regards. I recommend publication in its present form, save for two minor suggestions described below.

Author Response: Thank you; we appreciate this endorsement.

What are the major claims of the paper?

The authors have discovered much improved methods to merge positron and antiproton plasmas to form trappable antihydrogen. Not only is the trapping efficiency improved, but trapped antihydrogen pulses can be "stacked," Thus, subsequent measurements on them can benefit from larger numbers of detected antiprotons and hence increased signal to noise relative to using the individual pulses serially.

Are they novel and will they be of interest to others in the community and the wider field?

These new techniques include several substantive new methods and protocols. The increased trappable antiatom production rate is increased about an order of magnitude. This is a huge improvement over previous work. Many groups are either studying now, or plan to study, antihydrogen. The present work should be of considerable interest to those focusing on antihydrogen, and more generally, to those interested in improved techniques to create tailored collections of antiparticles.

Is the work convincing, and if not, what further evidence would be required to strengthen the conclusions?

The work appears to be carefully done and the manuscript is well written.

On a more subjective note, do you feel that the paper will influence thinking in the field?

The answer is definitely "yes."

Please feel free to raise any further questions and concerns about the paper.

Two minor comments. In Line 207, I have trouble determining what "the latter" refers to. It might be good to rephrase this passage. The sentence on lines 213 - 218 would benefit from rewriting also.

Author Response: Both passages have been rewritten for clarity.

We would also be grateful if you could comment on the appropriateness and validity of any statistical analysis, as well the ability of a researcher to reproduce the work, given the level of detail provided.

1. The statistical analysis is good.
2. The paper makes clear what the new procedures are, and so it is certainly adequate in that regard. As mentioned above, the techniques described will, for example, be of rather direct use to the other antihydrogen groups at CERN.

Reviewer #2 (Remarks to the Author):

Dear Editor,

the manuscript

Antihydrogen accumulation for fundamental symmetry tests

by the ALPHA collaboration of CERN reports on two new achievements in antihydrogen physics,

- 1.) a tenfold improved antihydrogen trapping rate, compared to earlier experiments, and
- 2.) the first demonstration of stacking of antihydrogen atoms and the report of the detection of 54 antihydrogen annihilations from a single trap release.

Both achievements together constitute a significant and very important improvement in antihydrogen research and are crucial ingredients to reach the final goals of this branch of fundamental experimental physics – the high-precision comparisons of the properties of hydrogen and antihydrogen to test the fundamental CPT invariance of the Standard Model of Particle Physics.

My overall impression on this manuscript is very positive and given the described results, which considerably contributed to the recent great success of the ALPHA collaboration, publication in Nature Communications can be considered after revision.

Author Response: Thank you; we appreciate this endorsement.

The manuscript details procedures which were applied to increase the antihydrogen trapping rate in ALPHA. An important aspect which is missing, is the discussion of the statistical stability of the results. It is obvious, e.g. from the linear scaling of the stacking experiments, that the stability of the experimental sequences meets the requirements which allow the conclusions made in the manuscript, but the discussion of the run-stability should be much more explicit. E.g. just by giving the number of runs which contribute to the results shown in Tab. 1, together with the errors and the explicit statement that the given error is based on the assumption that the results come from a specified parent distribution would already be convincing.

Author Response: We have modified Table 1 and the associated caption, as suggested.

A second point which I would like to address is that the details of the determination of many parameters which enter this manuscript are not given. One important example is the positron temperature, a parameter which is used in the qualitative interpretation of the data. I'm aware that plasma temperature measurements are described elsewhere, but given the importance of positron temperature for this manuscript, it would be adequate to briefly comment for example: "Positron temperature measurements are described in [refs]. For the temperature values used in this manuscript we apply the following method...[brief but explicit description of the experiment sequence]..."

Author Response: This information was in fact already specified in lines 120-121 of the original manuscript. We have elaborated on the description and have added yet another reference, in the spirit of the Reviewer's suggestion.

One of the major data sets which is presented in this paper is summarized in Tab 1. As argued above, a column should be added which displays the number of runs taken for each parameter-set. All the parameters contain errors except the amount of injected pbars, the authors may want to comment on that or may want to add an error to the pbar number. The table caption summarizes the positron densities, obviously different densities were chosen for the AR experiments and the potential merge experiments. Naively I would have expected that higher positron densities would lead to even higher hbar production rates. Can the authors please comment why they have chosen lower positron densities in the potential merge schemes? How does this parameter change/affect the direct comparison of the two schemes? Is there any theoretical model which would consistently describe the AR and the PM dataset just by changing positron density and positron temperature? No problem if not, but then the authors should add a short comment. It is not necessarily required, but if the results could be presented in a more theoretical rather than a purely qualitative framework, it would add additional quality to the manuscript.

Author Response: The table has been modified precisely as suggested.

As for the fact that the potential merge scheme outperforms AR mixing when lower positron densities are chosen, we unfortunately do not yet have a satisfactory theoretical model. Moreover, our data are limited in scope, and so any discussion of this point would necessarily be lengthy and unsubstantiated. We feel it is prudent to simply state the conditions that yield the observed improvements. This is clearly an issue that will need to be unravelled down the road.

In the spirit of the reviewer's queries, we have added a statement to the caption of Table 1 pointing out that the initial conditions we selected were chosen empirically so as to optimize the trapping efficiency.

Throughout the description of the experiment sequences detailed time information is missing. Only P6/L226 summarizes that the above described preparation sequence takes about 4 minutes. The authors should provide more detailed timing information along the entire description of the experiment sequences and I even propose to add a figure which illustrates the timing / parallelization sequence. This will support the readability of the manuscript. How long are e.g. compression times, ramp times for evaporative and adiabatic cooling etc..? I'm aware that these results were partly published elsewhere, but to make this manuscript more self-consistent, it would be important to describe the exact sequence which was applied to produce these results in a more detailed way.

Author Response: We have added considerably more timing information to the manuscript as requested, but feel that elaborating to the extent suggested by the reviewer would obscure the essential findings we aim to report. In particular, we wish to avoid overloading the reader with information that would not likely port over to another apparatus with different boundary conditions and operating constraints. Most of the timings that we do employ are tuned in situ, and represent experimental compromises between many factors that are unlikely to

translate to other experiments, and thus we have no reason that these times are essential to the findings reported.

Once these comments will be addressed in a revised version of the manuscript, I will recommend publication in NC.

Detailed Comments:

Page 2, Line 63 ff:

For a journal like NC, I propose that the authors provide a broader introduction. Rather than just describing CPT tests with antihydrogen also other CPT-testing efforts (antiprotons / antiprotonic Helium / muons / electron-positron) should be summarized/cited in one or two sentences.

Author Response: We have added the requested information, including additional references, to the second paragraph of the manuscript.

Page 3, Line 112 ff:

Description of pbar cooling by electrons: "Most" of the electrons are ejected before RW cooling, and afterwards the remaining fraction of electrons is removed. I'd like to ask the authors to detail: Which fraction of electrons is initially removed, and how long is the cooling time after RW compression before the remaining electrons are ejected? The quantification of the cycle time of such a preparation cycle would be interesting.

Author Response: The requested information has been added.

Page 3, Line 126 ff:

Description of re-preparation of pbars in ALPHA2: Please add also here time information. How long does the re-preparation take, and how long are your evaporative cooling ramps?

Author Response: The positron cooling ramp time is already stated in the caption to Figure 2. The pbar evaporation ramp time (10 s) has been added to the text.

Page 4, line 140 ff:

"which can vary from the accumulator"

About which variations are we talking here? 1%, 10% or even 50% to 90%? Please specify. If possible, please indicate how this variation affects the production/trapping rate of hbars, and how the fluctuations influence the scatter given in Tab 1.

Author Response: Information to this effect has been added to the manuscript. Once controlled, residual variations are at the 1 % level. (A manuscript describing the methods used to establish this control is being prepared, and will be reported elsewhere.)

We have not established a correlation between these fluctuations and hbar trapping/production rates, and are thus not able to respond further without venturing into the realm of speculation.

Page 4 / Page 5, Lines 173 to 187:

Very interesting and nice description. Although already published elsewhere, it would improve the self-consistency and readability of this article and add one or two sentences which describe how positron temperature is measured. Please comment on the significance of the temperature measurements. E.g. if the amount of positrons during the potential merge scheme is reduced, how is their temperature determined after a mixing process? Or is this temperature information based on offline experiments without antiprotons, or is a "leak rate" measured online while the potential merge scheme is performed?

Author Response: The requested information was already stated on lines 120 and 121 of the original manuscript. Note however that we have elaborated on the original description, and now also point out that the positron temperature is measured at the end of each mixing cycle, as part of the process of emptying the trap of charged particles.

Citations:

Page 2, Line 91: "All experimental initiatives in this field..." -> other antihydrogen collaborations could be cited.

Author Response: We have added references at this point to work published by two other collaborations that produce antihydrogen using similar techniques as well as to a review of antihydrogen physics.

Reviewer #3 (Remarks to the Author):

The article "Antihydrogen accumulation for fundamental symmetry tests" by Ahmadi et al (ALPHA Collaboration) reports on several advances in techniques useful for antihydrogen production and trapping, which in summary lead to a tenfold increase of the antihydrogen trapping efficiency. This work is very useful in the context of upcoming precision spectroscopy of antihydrogen, as a test of the fundamental principles of Nature.

Among the reported improvements, two are particularly noteworthy. The "potential merge mixing" of the positron and antiproton plasmas yields, somewhat counterintuitively, significantly more, and also colder antihydrogen. Its yield is also scalable by increasing the number of trapped antiprotons. Thus this new technique will be of prime importance to exploit the 100x larger antiproton plasmas expected at the new ELENA facility.

The second new technique is the demonstration of "stacking" of antihydrogen atoms from several successive mixing cycles. A linear scaling in the number of detected antiprotons is observed, paving the way to high precision high signal-to-background spectroscopy.

The manuscript is very well written and the few comments I have are really miniscule:

Author Response: Thank you; We appreciate these kind words.

line 77: "low enough" could be quantified, (0.5 K for the 1 Tesla field employed). This number is given only on page 5 (line 205)

Author Response: We have added the requested information.

line 105: A word on why "only" 73% of the antiprotons are cooled may be in order. What prevents the remaining ones from being cooled?

Author Response: In principle, nothing prevents the remaining antiprotons from being cooled. The fraction of antiprotons cooled depends on the number and radial extent of the electron plasma used, as described in Ref. 11 (now 15). To facilitate the remainder of the preparation process, including making the entire sequence fit within the AD cycle, the current implementation of the electron preparation results in 73% cooling efficiency. In principle, all that is required to increase this to 100% is sequence development time. However, sequence development time comes at a premium (in the sense that beam time from the AD is a precious resource) and it turned out to be a non-trivial exercise to increase this fraction further and at the same time make the protocol compatible with the rest of the preparation sequence.

We have added a statement to the manuscript and a new citation of Ref. 11 (now 15) to explain this technical detail, and point out that it is not a fundamental limitation.

line 106,118: The authors switch between eV and K. The broader readership would probably benefit from mentioning a conversion factor or giving once both the energy and the temperature.

Author Response: Good point. A clarifying statement has been added.

REVIEWERS' COMMENTS:

Reviewer #2 (Remarks to the Author):

The authors have addressed all comments of my initial review and have replied to the raised questions in a satisfactory way.

I'm happy to recommend publication of this manuscript in Nature Communications and congratulate the ALPHA collaboration to another excellent manuscript.

Reviewer #3 (Remarks to the Author):

I'm happy. Please publish!

Thanks.